# Identity Leadership, Employee Burnout and the Mediating Role of Team Identification: Evidence from the Global Identity Leadership Development Project

**DOI:** 10.3390/ijerph182212081

**Published:** 2021-11-17

**Authors:** Rolf van Dick, Berrit L. Cordes, Jérémy E. Lemoine, Niklas K. Steffens, S. Alexander Haslam, Serap Arslan Akfirat, Christine Joy A. Ballada, Tahir Bazarov, John Jamir Benzon R. Aruta, Lorenzo Avanzi, Ali Ahmad Bodla, Aldijana Bunjak, Matej Černe, Kitty B. Dumont, Charlotte M. Edelmann, Olga Epitropaki, Katrien Fransen, Cristina García-Ael, Steffen Giessner, Ilka H. Gleibs, Dorota Godlewska-Werner, Roberto González, Ronit Kark, Ana Laguia Gonzalez, Hodar Lam, Jukka Lipponen, Anna Lupina-Wegener, Yannis Markovits, Mazlan Maskor, Fernando Molero, Lucas Monzani, Juan A. Moriano Leon, Pedro Neves, Gábor Orosz, Diwakar Pandey, Sylwiusz Retowski, Christine Roland-Lévy, Adil Samekin, Sebastian Schuh, Tomoki Sekiguchi, Lynda Jiwen Song, Joana Story, Jeroen Stouten, Lilia Sultanova, Srinivasan Tatachari, Daniel Valdenegro, Lisanne van Bunderen, Dina Van Dijk, Sut I. Wong, Farida Youssef, Xin-an Zhang, Rudolf Kerschreiter

**Affiliations:** 1Institute of Psychology, Goethe University, 60323 Frankfurt, Germany; 2School of Psychology, University of East London, London E15 4LZ, UK; j.lemoine@uel.ac.uk; 3ESCP Business School, London NW3 7BG, UK; 4School of Psychology, University of Queensland, St Lucia 4072, Australia; n.steffens@uq.edu.au (N.K.S.); a.haslam@uq.edu.au (S.A.H.); m.maskor@uq.net.au (M.M.); 5Psychology Department, Dokuz Eylul University, Izmir 35390, Turkey; serap.akfirat@deu.edu.tr; 6Department of Counseling and Educational Psychology, De La Salle University, Manila 1004, Philippines; christine.ballada@dlsu.edu.ph (C.J.A.B.); aruta_johnjamirbenzon@yahoo.com (J.J.B.R.A.); 7Department of Social Psychology, Faculty of Psychology, Lomonosov Moscow State University, 125009 Moscow, Russia; tbazarov@mail.ru; 8Department of Psychology and Cognitive Science, University of Trento, 38068 Rovereto, Italy; Lorenzo.avanzi@unitn.it; 9Faculty of Business and Management, Information Technology University, Lahore 54700, Pakistan; bodla786@hotmail.com; 10Institute for Leadership and HRM, University of St. Gallen, 9000 St. Gallen, Switzerland; aldijana.bunjak@unisg.ch; 11School of Economics and Business, University of Ljubljana, 1000 Ljubljana, Slovenia; matej.cerne@ef.uni-lj.si; 12Department of Psychology, University of South Africa, Pretoria 0003, South Africa; dumonkb@unisa.ac.za; 13Department of Movement Sciences, KU Leuven, 3000 Leuven, Belgium; charlotte.edelmann@kuleuven.be (C.M.E.); katrien.fransen@kuleuven.be (K.F.); 14Durham University Business School, Durham University, Durham DH1 3LB, UK; olga.epitropaki2@durham.ac.uk; 15Universidad Nacional de Educación a Distancia, 28040 Madrid, Spain; cgarciaael@psi.uned.es (C.G.-A.); aglaguia@psi.uned.es (A.L.G.); fmolero@psi.uned.es (F.M.); jamoriano@psi.uned.es (J.A.M.L.); 16Rotterdam School of Management, Erasmus University, 3062PA Rotterdam, The Netherlands; sgiessner@rsm.nl; 17Department of Psychological and Behavioural Science, London School of Economics, London WC2 2AE, UK; I.H.Gleibs@lse.ac.uk; 18Institute of Psychology, University of Gdańsk, 80-309 Gdańsk, Poland; dorota.godlewska-werner@ug.edu.pl; 19School of Psychology, Pontificia Universidad Católica de Chile, Santiago 7820436, Chile; rgonzale@uc.cl; 20Department of Psychology and the Graduate Gender Studies Program, Bar-Ilan University, Ramat-Gan 5290002, Israel; karkronit@gmail.com; 21Business School, University of Exeter, Exeter EX4 4QG, UK; 22Department of Psychology, University of Amsterdam, 1001 NK Amsterdam, The Netherlands; h.w.lam@uva.nl (H.L.); l.vanbunderen@uva.nl (L.v.B.); 23Faculty of Social Sciences, University of Helsinki, 00014 Helsinki, Finland; jukka.lipponen@helsinki.fi; 24School of Engineering and Management Vaud, HES-SO, 1401 Yverdon-les-Bains, Switzerland; anna.lupina-wegener@heig-vd.ch; 25Independent Authority of Public Revenue, 57001 Thessaloniki, Greece; i.markovits@yahoo.com; 26Ivey Business School, Western University, London, ON N6G 0N1, Canada; lmonzani@ivey.ca; 27Nova School of Business & Economics, Carcavelos, 2775-405 Lisbon, Portugal; pedro.neves@novasbe.pt; 28Pluridisciplinary Research Unit: Sport, Health and Society (URL), Sherpas, Universite Artois, 62800 Lievin, France; gaborosz@gmail.com; 29Faculty of Management, Bhairahawa Multiple Campus, Tribhuvan University, Rupandehi 32900, Nepal; pandeydiwakar22@gmail.com; 30Faculty of Psychology, SWPS University of Social Sciences and Humanities, 81-745 Sopot, Poland; sretowski@swps.edu.pl; 31Department of Psychology, University of Reims Champagne-Ardenne, 51097 Reims, France; Christine.Roland-Levy@univ-reims.fr; 32School of Liberal Arts, M. Narikbayev KAZGUU University, Nur-Sultan 010000, Kazakhstan; s_adil@list.ru; 33Department of Organizational Behavior and Human Resource Management, China Europe International Business School (CEIBS), Shanghai 201206, China; sschuh@ceibs.edu; 34Graduate School of Management, Kyoto University, Kyoto 606-8501, Japan; tomoki@econ.kyoto-u.ac.jp; 35Leeds University Business School, University of Leeds, Leeds LS2 9JT, UK; L.Song@leeds.ac.uk; 36Sao Paulo School of Business Administration, Fundação Getulio Vargas, Sao Paulo 01313-902, Brazil; joana.story@fgv.br; 37Faculty of Psychology and Educational Sciences, KU Leuven, 3000 Leuven, Belgium; jeroen.stouten@kuleuven.be; 38Department of Psychology, Lomonosov Moscow State University, Branch of Tashkent, Tashkent 100060, Uzbekistan; liks78@mail.ru; 39Human Resources, Organizational Behaviour and Communications Area, T A Pai Management Institute, Karnataka 5761004, India; srinivasant@tapmi.edu.in; 40School of Politics and International Studies, University of Leeds, Leeds LS2 9JT, UK; dhvalden@gmail.com; 41Department of Health Policy and Management, Ben-Gurion University of the Negev, Be’er Sheva 8410501, Israel; dinav@bgu.ac.il; 42Department of Communication and Culture, BI Norwegian Business School, 0484 Oslo, Norway; sut.i.wong@bi.no; 43Cairo Institute of Liberal Arts and Science, Giza 12311, Egypt; fyoussef93@aucegypt.edu; 44Antai College of Economics and Management, Jiao Tong University, Shanghai 200030, China; xinanzhang@sjtu.edu.cn; 45Department of Education and Psychology, Freie Universität Berlin, 14195 Berlin, Germany; rudolf.kerschreiter@fu-berlin.de

**Keywords:** burnout, exhaustion, identity leadership, team identification, cross-cultural study

## Abstract

Do leaders who build a sense of shared social identity in their teams thereby protect them from the adverse effects of workplace stress? This is a question that the present paper explores by testing the hypothesis that identity leadership contributes to stronger team identification among employees and, through this, is associated with reduced burnout. We tested this model with unique datasets from the Global Identity Leadership Development (GILD) project with participants from all inhabited continents. We compared two datasets from 2016/2017 (*n* = 5290; 20 countries) and 2020/2021 (*n* = 7294; 28 countries) and found very similar levels of identity leadership, team identification and burnout across the five years. An inspection of the 2020/2021 data at the onset of and later in the COVID-19 pandemic showed stable identity leadership levels and slightly higher levels of both burnout and team identification. Supporting our hypotheses, we found almost identical indirect effects (2016/2017, *b* = −0.132; 2020/2021, *b* = −0.133) across the five-year span in both datasets. Using a subset of *n* = 111 German participants surveyed over two waves, we found the indirect effect confirmed over time with identity leadership (at T1) predicting team identification and, in turn, burnout, three months later. Finally, we explored whether there could be a “too-much-of-a-good-thing” effect for identity leadership. Speaking against this, we found a u-shaped quadratic effect whereby ratings of identity leadership at the upper end of the distribution were related to even stronger team identification and a stronger indirect effect on reduced burnout.

## 1. Introduction

Burnout is a widespread phenomenon that affects employees across a variety of professions and which has been observed—and studied—around the world. Regarding the scale of the problem, a recent study of over 20,000 health care employees found that up to 49% of respondents reported suffering from burnout [1]. In the present paper, we seek to contribute to the understanding of this issue by exploring the capacity for leadership to protect employees from job burnout. More specifically, we propose that identity leadership, which helps to create a sense of shared identity within a team, helps to build and sustain team identification, thereby reducing team members’ experience of exhaustion. To test this proposition, we examine these relationships in a large dataset comprised of samples from 28 different countries.

Job burnout has been conceptualized by Maslach and others [2,3] as having three core dimensions—namely, emotional exhaustion, feelings of reduced personal accomplishment and depersonalization. Of these, emotional exhaustion is arguably the core component and it is certainly the one that has been subjected to most empirical investigations [4]. Here, the meta-analyses found that burnout is associated with reduced self-efficacy at work [5] and with conflicts between the work and non-work domains [6]. Another meta-analysis by Aronsson et al. [7], that summarized the results of 25 longitudinal studies, found that employees’ experience of justice and support at work helped to protect them against emotional exhaustion, whereas high demands, low job control, high workload, low reward and job insecurity increased their risk of suffering from exhaustion. Indeed, it was partly for this reason that, in 2019, the World Health Organization moved to include burnout within their International Classification of Diseases [8].

In sum, then, there is large body of research which suggests that burnout is a widespread phenomenon that has negative effects on both the individual and the organization. This is also understood to be caused by poor working conditions [9]. Importantly, though, there is also evidence that these conditions—and workers’ experience of them—can be heavily structured by line managers and team leaders, not least through their provision of social support [7].

### 1.1. Social Identification and Burnout

In regard to these various issues, a growing body of research shows that health in the workplace is affected by the sense of identity that employees derive from their membership in social groups (i.e., their *social identity*) [10]. In particular, social identity researchers have argued that people’s social identities are a psychological resource and that they have important consequences for health [11,12]. This is because, among other things, social identity is a basis for (a) the provision and receipt of social support [13], (b) a sense of connection to others [14], (c) a sense of control [15], (d) a sense of collective self-efficacy [16] and (e) a sense of meaning and purpose [17]. These processes in turn are also argued to minimize—and to help people work together to counteract—the harmful effects of various stressors they encounter in the workplace in ways that protect them from burnout [18,19].

A number of previous studies has tested these ideas by exploring the relationship between social identity and the development of burnout [12]. One of the first to do so was an in-depth analysis of participants’ stress trajectories in the BBC Prison Study [19]. Here, over the course of six days, the prisoners developed a sense of a shared social identity and supported each other in challenging the guards, while, in the face of this confrontation, the guards’ identification with their group declined. Hand in hand with their declining identification the guards also reported higher burnout as the study progressed, such that, by Day 6, they were significantly more burnt-out than the guards. In another longitudinal study, Haslam et al. [20] surveyed members of a theatre production team at various stages of the production (after audition, at dress rehearsal, before and after the final production) and found that those who were more strongly identified with the team were less likely to suffer from burnout—especially at critical phases of the production.

Other organizational research studies found similar patterns that point to the protective role of social identification. For example, Avanzi et al. [21] surveyed over 2500 Swiss teachers and found a negative relationship between organizational identification and burnout. This was mediated by both increased social support and perceptions of reduced workload. Along similar lines, a study of Italian high school teachers by Avanzi et al. [16] supported a mediation model in which organizational identification was associated with lower burnout via increased social support and higher collective self-efficacy (for experimental evidence, see also [22], Study 2). The robust nature of the negative relationship between social identification and burnout was also confirmed in a meta-analysis by Steffens and colleagues (*k* = 58) [23]. Looking at the relationship between both organizational and team identification and indicators of physical and psychological well-being, this study found that both forms of social identification were reliably associated with the absence of stress (of which burnout was typically a component).

Nevertheless, a number of studies has also failed to establish a direct link between organizational identification and burnout. For instance, although, as expected, the relationship was negative (*r* = −0.12), Ciampa et al. did not find a significant direct link between employees’ organizational identification and their exhaustion [24]. Instead, they found that this relationship was contingent on employees’ ambivalent identification with their organization, so that the expected negative relationship between identification and exhaustion was only apparent for employees with low ambivalence. The reasons for this are unclear, but one might imagine that leaders play a critical role here—not only in creating a sense of shared social identity with their team but also in reducing team members’ ambivalence.

### 1.2. Leadership and Burnout

Clear evidence of the importance of leadership for team members’ mental health is provided by Kuoppala and colleagues’ meta-analysis (*k* = 27) of the relationship between leadership and burnout [25]. This found that burnout was negatively associated (the authors calculated risk ratios (RR) with three key aspects (or forms) of leadership: consideration (RR = 1.85), supportive leadership (RR = 1.32) and transformational leadership (RR = 1.95). More recently, Harms et al. conducted a meta-analysis to explore the relationship between burnout and a slightly different set of leadership constructs: transformational leadership, leader–member exchange and abusive supervision [26]. Again, they found that these aspects and forms of leadership were significant predictors of employee burnout (*k* = 25, *r* = −0.32; *k* = 18, *r* = −0.45; *k* = 13, *r* = 0.22, respectively).

Of note here, the leadership constructs that were examined in the primary studies on which the above two meta-analyses were conducted typically conceptualized the leader as someone who was in an exalted position rather than a core member of the groups they led. In contrast, Haslam and colleagues argued that, if leaders are set apart from the group, this often compromises their leadership [27]. This, they argued, is because leadership is a process in which leaders are effective by gaining *power through* followers rather than by wielding *power over* them [28]. Haslam and colleagues [29] expand upon these ideas by setting out a “new psychology of leadership”, which argues that leaders’ effectiveness rests on their capacity to build and advance a sense of shared social identity (a sense of “us-ness”) with those they are seeking to influence and to motivate—through a process they refer to as *identity leadership* [30,31]. As they set it out, identity leadership has four key components: (a) *identity prototypicality*, whereby a leader is seen to embody a sense of shared social identity as “one of us” [32]; (b) *identity advancement*, whereby a leader promotes and defends the group’s collective interests (rather than their personal interests or those of other groups) and so is “doing it for us” [33]; (c) *identity entrepreneurship*, whereby a leader works to cultivate a sense of shared identity and so is seen to be “crafting a sense of us” [34,35]; (d) *identity impresarioship*, whereby a leader works to translate social identity into material reality by initiating structures, activities, events and rituals that allow group members to come together in ways that are seen to be “making us matter” [27]. Together, these four aspects of identity leadership help team members to identify more strongly with their team members and, as a result, motivate them to display the engaged followership that translates the leader’s vision into action in the world [29].

### 1.3. Identity Leadership, Team Identification and Burnout

To date, most of the research works that have been inspired by the social identity approach to leadership have focused on the first dimension of identity leadership, the leader’s group prototypicality. Here, meta-analyses have shown that leaders who are prototypical of the group they lead are not only more favorably evaluated and more trusted (*k* = 35) [36], but also more likely to create teams that are seen by their members as cohesive, high performing and supporting well-being (*k* = 128) [37]. However, group members’ well-being has also been found to be associated with identity entrepreneurship. Specifically, Steffens et al. surveyed over 600 employees in the US and found that team members’ perceptions of identity entrepreneurship predicted lower burnout (which, in turn, predicted better-perceived team performance) [38]. More recently, in a sample of 363 German employees, Krug et al. also found that leaders’ identity entrepreneurship predicted well-being during the COVID-19 pandemic—specifically in the form of reduced burnout and loneliness [39]. Van Dick et al., in a survey of employees across 20 countries, also found negative correlations between identity leadership, its four components and burnout; a simultaneous regression analysis showed that identity advancement was the strongest predictor of burnout [30]. In another study of 854 Spanish employees [40], Laguía and colleagues found identity entrepreneurship to be positively related to positive affect and negatively related to negative affect and both types of affects, in turn, related to work engagement. In the domain of sports, Fransen et al. conducted a survey study of 289 handball players and found that, when they perceived their coaches, captains and informal leaders to be strong in identity leadership, they identified more with their teams, which, in turn, increased feelings of psychological safety, which was then negatively related with burnout [41]. Finally, Steffens et al. found identity leadership to be related to team identification and job satisfaction in a sample of 699 US employees [31] and simultaneous regressions revealed that identity prototypicality and identity advancement predicted job satisfaction, while identity prototypicality, identity entrepreneurship and identity impresarioship predicted team identification.

Pulling the various strands of the foregoing review together, we see that previous research studies provide support for three key propositions. First, it is clear that leaders can be a source of team members’ burnout. Second, a sense of shared identity in a team is likely to have positive impact on its members’ well-being—in particular, by increasing social support and collective self-efficacy. Third, leaders’ identity leadership is likely to foster team members’ team identification and this, in turn, should contribute to those team members’ well-being.

In line with these ideas, Krug et al. conducted a survey of 192 German employees and found that the team leaders’ perceived identity leadership was associated with higher team member identification and, through this, with lower burnout [42]. However, this study was limited by the fact that it had a cross-sectional design and data were obtained in only one cultural context. Therefore, the present research project seeks to provide a more robust test of these propositions by utilizing data collected in a very broad range of cultural contexts and also analyzing a subset of this dataset that surveyed participants at two waves.

### 1.4. The Present Research Project

In line with the propositions set out above, our research study sought to test the following hypotheses.

**Hypotheses** **1.**
*Team members’ perceptions of their supervisors’ identity leadership is associated with those team members identifying more highly with their team.*


**Hypotheses** **2.**
*We expect a negative indirect effect of team members’ perceptions of their supervisors’ identity leadership with team members’ burnout, via team identification.*


As well as testing these a priori hypotheses, we also seek to leverage a large and culturally diverse dataset to examine three additional research questions which this study is particularly well suited to address. First, the fact that our data were obtained in identical ways (i.e., using the same methods of data collection and identical questionnaire instruments) in two related projects that were conducted before and during the COVID-19 pandemic (in 2016/2017 and 2020/2021) allows us to examine the degree to which identity leadership, team identification and burnout change across time and in the context of this unprecedented global threat to health (RQ1).

Second, whilst a large number of studies of identity leadership, team identification and burnout have been conducted in many different countries, there has previously been no integrated approach that uses the same methodology at the same time in a way that would allow us to know whether and to what degree our hypothesized relationships are supported across cultural contexts. In this regard, our dataset is unique, in having been obtained from employees on all inhabited continents. Therefore, this allows us to compare support for H1 and H2 across the eight cultural clusters previously identified by the GLOBE research program [43] (RQ2). Testing for stabilities or differences across cultures is important, as there is some previous evidence of cultural difference in the effects of identification between cultures. Lee and colleagues, in a large meta-analysis with over 114 studies, found stronger relationships between organizational identification and work-related attitudes and behaviors in collectivistic cultures (compared to individualistic cultures), but they did not find any other influences of uncertainty avoidance or long-term orientation [44].

Third, in their research study on the relationship between team identification and team functioning, Avanzi and colleagues found evidence of a “too-much-of-a-good-thing” effect if employees identified very highly with their teams [45,46]. More specifically, across three studies, they observed curvilinear relationships between organizational identification and employee health (including, in Study 3, employee exhaustion). In line with Steffens et al.’s meta-analytic findings [23] and the broader social cure literature [12] in all three studies, moderate levels of identification were associated with better health than low levels of identification. However, when identification was very high, employees reported poorer health than when it was moderately high. Avanzi et al. attributed this to workaholism and overcommitment [45,46] on the part of those whose personal identities were fused with the social identity of their team [47]. In the present research project, we are in a position to test whether identity leadership might have the same negative impact if leaders take identity-building activities to extremes (RQ3). To be clear, we do not expect such curvilinear effects of identity leadership, as we believe that there is no threshold of turning too much good leadership into negative effects. However, in the spirit of open mindedness as one of the underlying principles of good science, we put RQ3 to a test in an exploratory way.

In the context of testing H1 and H2, our study seeks to explore RQ1, RQ2 and RQ3 using two datasets collected specifically for this purpose. One is an international study of more than 7000 employees from 28 countries, the other a subset of 111 German participants who completed the survey again 12 weeks later. We use these datasets to test both H1 and H2 and the international dataset to explore cultural differences and potential curvilinear effects. In addition, we use a dataset collected five years earlier to examine the stability of the constructs over time.

## 2. Materials and Methods

### 2.1. Sample

The *GILD—Global Identity Leadership Development*—collaboration project comprises a guild of international researchers in the field of social and organizational psychology. The project started in 2016 and the first phase of data collection was completed in 2017. In the present paper, we describe the second phase of data collection with a modified questionnaire. Surveys were coordinated and managed mainly by the first five and the last author of this manuscript and distributed by the entire team of researchers in 28 countries using snowball techniques.

Participants took part voluntarily; the surveys were anonymous and respondents could interrupt their participation at any time without any consequences. Researchers in each country attempted to collect data from at least 200 participants in 2020 and 2021. This was achieved in 19 countries: Australia (*n* = 269), Belgium (*n* = 285), Bosnia and Herzegovina (*n* = 241), Brazil (*n* = 222), Canada (*n* = 353), China (*n* = 445), Germany (*n* = 554), Greece (*n* = 210), Israel (*n* = 215), Japan (*n* = 284), the Netherlands (*n* = 270), Norway (*n* = 200), the Philippines (*n* = 281), Poland (*n* = 375), Portugal (*n* = 202), Spain (*n* = 692), Switzerland (*n* = 216), United Kingdom (*n* = 263) and United States (*n* = 318). In 8 other countries, researchers collected data from slightly fewer participants: France (*n* = 123), India (*n* = 192), Italy (*n* = 191), Kazakhstan (*n* = 161), Pakistan (*n* = 172), Russia (*n* = 171), Turkey (*n* = 190), Uzbekistan (*n* = 103) and, with slightly less than 100 participants, Slovenia (*n* = 96).

Therefore, the final dataset consisted of 7294 participants from 28 countries and 31 regions as Switzerland and Pakistan collected data in more than one language in different parts of the country. The countries were categorized into eight clusters (in line with previous GLOBE research projects) [43]: Anglo (Australia, United States, Canada and United Kingdom), Confucian Asia (China and Japan), Eastern Europe (Greece, Poland, Bosnia and Herzegovina, Slovenia, Russia, Uzbekistan and Kazakhstan), Germanic Europe (Belgium, the Netherlands and Germany), Latin America (Brazil), Latin Europe (France, Italy, Portugal, Switzerland, Israel and Spain), Nordic Europe (Norway) and Southern Asia (Turkey, India, Pakistan and the Philippines). The English master survey was translated (using the translation-back-translation method) [48] into 19 different languages. Table 1 provides an overview of the characteristics of the total sample and of the sample from each country.

Participants worked in both the private and public sector and across different industries. They were heterogenous in their age, work experience (in general and in their current company) and gender (see Table 1 for details). They worked for companies with an average of 8631 employees (*SD* = 50,197; range, 1–1,000,000; median = 180) and in teams with an average of 14.76 employees (*SD* = 16.80; range = 1–149; median = 9, excluding fewer than 1% of participants who reported having teams with more than 150 members).

### 2.2. Time Span

After the survey was successfully piloted in Poland from November 2019 to January 2020, data collection started in other countries from February 2020 and lasted until May 2021. We clustered the countries into those in which data were collected in the early stages of the COVID-19 pandemic (i.e., from February 2020 to June 2020: China, India, Israel, Japan, the Netherlands, Norway, the Philippines and the United Kingdom) and those in which data collection took place later in the crisis (i.e., from September 2020 to May 2021: Australia, Belgium, Canada, Italy, Portugal, Russia, Switzerland, Turkey and the United States). In the remaining 11 countries, data collection took place over a longer period that spanned both time intervals.

From November 2020 to January 2021, we collected data in Germany and asked participants (*n* = 111) to participate in a follow-up survey twelve weeks later (from February 2021 to April 2021). Participants were invited via email and received a reminder about two weeks after the invitation. To match the surveys, participants created a personalized code comprised of letters and numbers. Of this smaller sample, 73.9% were female and age was uniformly distributed (at the first measurement point, 9% were 18–25 years; 36.9%, 26–35, years; 27%, 36–45; 19.8%, 46–55 years; 7.2%, over 55 years).

The composition of countries was comparable to the first GILD dataset collected in 2016/2017 (*n* = 5.290; 20 countries); see [30]. In 13 countries, data were gathered in both project phases, allowing patterns to be compared across the two time points (Australia, Belgium, China, France, Germany, Greece, India, Israel, Italy, Japan, the Netherlands, Norway and Turkey).

### 2.3. Measures

The co-authors of this paper translated the English survey into their native language for each of the 21 countries where English was not the native tongue. If available, the translations of the relevant scales in the first phase of GILD [30] were used. We used the back-translation method suggested by Brislin [48] and inconsistencies were discussed before agreeing on the best possible solution. Translated items from the ILI scales are provided in the Appendix A.

The 15-item *Identity Leadership Inventory* (ILI) developed by Steffens et al. [31] was used to measure the four dimensions of identity leadership: leader prototypicality (4 items, e.g., “My team leader exemplifies what it means to be a member of the team”), identity advancement (4 items, e.g., “My team leader acts as a champion for the team”), identity entrepreneurship (4 items, e.g., “My team leader creates a sense of cohesion within the team”) and identity impresarioship (3 items, e.g., “My team leader creates structures that are useful for team members”). Participants were instructed to think of their direct supervisor while responding to these items on 7-point scales (where 1 = “disagree completely”, 7 = “agree completely”).

*Team identification* was assessed using [49] a 4-item measure (e.g., “I consider myself to be part of my team”). Again, responses were made on 7-point scales (where 1 = “disagree completely”, 7 = “agree completely”).

*Burnout* was assessed using the 9-item emotional exhaustion subscale from Maslach and Jackson’s [3] burnout inventory (e.g., “I feel used up at the end of the work day”). Responses were made on a 7-point scale (where 1 = “never”, 7 = “every day”).

### 2.4. Analytic Procedure

Before proceeding with the main analyses, we tested all scales and items for invariance across countries. Unless stated otherwise, all of the following analyses were performed with the whole dataset. For the ILI scale, the factor loadings *R^2^* and intercepts *R^2^* were good and suggest a high level of invariance of the ILI. There were 1.7% of factor loadings that were not invariant and 22.4% of intercepts that were not invariant. Averaging the proportion of non-invariant factor loadings and intercepts, the total invariance of the ILI was 12.05%, which is below the 25% threshold [50]. For team identification, the factor loadings *R^2^* and intercepts *R^2^* were good and suggest a high level of invariance of the team identification scale. There were 0.8% of factor loadings that were not invariant and 10.8% of intercepts that were not invariant. Averaging the proportion of non-invariant factor loadings and intercepts, the total invariance of the team identification scale was 12.05%, which is below the 25% threshold. The only exception where we did not find invariance was the small subsample of 22 participants from Switzerland who answered the survey in English. For burnout, the factor loadings *R^2^* and intercepts *R^2^* were good and suggest a high level of invariance of the burnout scale. There were 5% of factor loadings that were not invariant and 47.7% of intercepts that were not invariant. Averaging the proportion of non-invariant factor loadings and intercepts, the total invariance of the burnout scale was 26.35%, which is just above the 25% threshold. This is mainly due to the small sample sizes of participants in Pakistan who answered in Urdu (*n* = 33) and those in Switzerland who answered in English (*n* = 22) and German (*n* = 30). For the longitudinal data from Germany, the factor loadings and intercepts were invariant for all items.

To test H1 and H2, we conducted a mediation analysis using the SPSS plug-in Process by Hayes [51], as well as the MEMORE (MEdiation and MOderation in REpeated-measures designs) calculation [52]. To explore RQ1, we compared the means of variables between the 2016/2017 and 2020/2021 samples and between early and later in the pandemic, by conducting independent sample *t*-tests using SPSS Version 26. To explore RQ2, we used a mediation analysis to compare support for H1 and H2 across the different GLOBE clusters. Finally, we used linear multiple regression analyses to test the predictive validity of the identity leadership dimensions with and without team identification in relation to burnout. To explore possible curvilinear effects (RQ3), a curvilinear analysis was conducted using Medcurve [53]. The bootstrapping analysis used 5000 resamples and 95% CIs.

## 3. Results

Participants who had more than 5% of missing values or who answered the survey in a very short time (less than eight minutes) were deleted from the dataset (*n* = 540, or 6.9%). We replaced the missing values in remaining responses using random imputation within the mice package [54]. In the overall dataset, 5234 values were imputed out of 714,812 values; therefore, they represent only 0.7%. Regarding the variables analyzed in this paper, 614 values were imputed out of 204,232 values, which only represents 0.3%. Re-analyses of the data without imputation revealed virtually identical results. In the German sample with two measurement time points, the responses from every participant who completed both surveys were analyzed. The inspection of skewness and kurtosis [55] showed that the three constructs ILI, team identification and burnout were normally distributed. ILI and team identification showed a negative skew, while burnout had a positive skew.

The inter-correlations of the entire sample between ILI and the four dimensions, team identification and burnout, as well as the reliability of the scales, are presented in Table 2. As can be seen from this table, all variables were significantly associated with each other but to a varying degree (all |*r*|s > 0.26). As can be seen, the reliabilities for the full dataset were excellent, with Cronbach’s alpha exceeding 0.90 in all cases. An inspection of the reliabilities for each country showed that there was only little variation (identity leadership had the lowest alpha in Pakistan, 0.95, and the highest alpha in the United States, 0.98; identity prototypicality had the lowest alpha in Pakistan, 0.82, and the highest alpha in the United States, 0.97; identity advancement had the lowest alpha in Pakistan, 0.85, and the highest alpha in Bosnia and Herzegovina, 0.97; identity entrepreneurship had the lowest alpha in Pakistan, 0.85, and the highest alpha in Norway, 0.97; identity impresarioship had the lowest alpha in Pakistan, 0.86, and the highest alpha in Norway, 0.95; team identification had the lowest alpha in Pakistan, 0.85, and the highest alpha in the United States, 0.95; burnout had the lowest alpha in Greece, 0.88, and the highest alpha in the United States, 0.97).

### 3.1. Testing H1 and H2 in the Cross-Sectional Sample

To test our main hypotheses, we calculated a mediation analysis (see Figure 1). We first found a significant and substantial correlation of *r* = 0.51 between identity leadership and team identification, confirming H1. H2 predicted an indirect effect from identity leadership to burnout via team identification and, in line with this hypothesis, we found a significant negative indirect effect for both the 2020/2021 and the 2016/2017 samples. The indirect effect was reliable for the full sample in 2020/2021 (*b* = −0.13; 95% CI between −0.15 and −0.12; see Figure 1) and virtually identical to the indirect effect of the 2016/2017 sample (*b* = −0.13; 95% CI between −0.14 and −0.12).

The negative indirect effects of identity leadership’s dimensions on burnout via team identification were also reliable in the 2020/2021 sample: identity prototypicality, *b* = −0.12, 95% CI between −0.13 and −0.11; identity advancement, *b* = −0.12, 95% CI between −0.13 and −0.11; identity entrepreneurship, *b* = −0.13, 95% CI between −0.14 and −0.11 and identity impresarioship, *b* = −0.12, 95% CI between −0.13 and −0.11.

### 3.2. Testing H1 and H2 in the Two-Wave Data

The German subsample allowed us to perform analyses of the relations between two time points 12 weeks apart. The inter-correlations between ILI, team identification and burnout, as well as the reliability of the scales at T1 and T2, are presented in Table 3.

Supporting H1, there was a reliable correlation between identity leadership at T1 and team identification at T2, *r* = 0.40. Supporting H2, a mediation analysis using PROCESS [51] (see Figure 2) showed that identity leadership at T1 significantly predicted burnout at T2 via team identification at T2: identity leadership: *b* = −0.10, 95% CI between −0.20 and −0.02; identity prototypicality: *b* = −0.09, 95% CI between −0.17 and −0.02; identity advancement: *b* = −0.10, 95% CI between −0.17 and −0.03; identity entrepreneurship: *b* = −0.10, 95% CI between −0.19 and −0.03; identity impresarioship: *b* = −0.09, 95% CI between −0.17 and −0.02. To confirm the results, we used MEMORE [48] and regressed the difference between team identification at T2 and T1 onto the difference between burnout at T2 and T1 and found a significant effect (*M*diff = −0.15, 95% CI between −0.29 and −0.01). This provides evidence of directionality, whereby changes in team identification over time contributed to changes in burnout at T2 [52].

### 3.3. Comparison across Time (RQ1)

To explore RQ1, we conducted independent sample *t*-tests to compare the means of our three focal constructs (identity leadership, team identification and burnout) across time in the 13 countries that participated in both waves (2016/2017 vs. 2020/2021). This revealed a significant but small increase in mean level of team identification (*M*_2016_ = 5.08, *SD*_2016_ = 1.47, *M*_2021_ = 5.25, *SD*_2021_ = 1.36, *t*(6655) = −4.97, *p* < 0.001, *d* = 0.12). However, there were no significant differences in the means for identity leadership or burnout (identity leadership: *M*_2016_ = 4.57, *SD*_2016_ = 1.57, *M*_2021_ = 4.64, *SD*_2021_ = 1.54, *t*(7043) = −1.96, *p* = 0.051, *d* = 0.05; burnout: *M*_2016_ = 3.20, *SD*_2016_ = 1.50, *M*_2021_ = 3.18, *SD*_2021_ = 1.43, *t*(7043) = 0.57, *p* = 0.568, *d* = 0.01). The results for each of the 13 countries are presented in Table 4. Within countries, there was a significant increase in team identification across the two time points in Belgium, Germany and Greece, but we observed a significant reduction in team identification in China. There was also a significant increase in identity leadership in France, Greece and Turkey, but a significant reduction in China and Germany. For burnout, we observed no differences other than in Japan, where there was a significant reduction over time.

We explored RQ1 further by comparing country data that were collected early (2020; *n_countries_* = 8) vs. later (2020–2021; *n_countries_* = 9) in the COVID-19 pandemic. This revealed no significant differences in identity leadership (*M_early_* = 4.48, *SD_early_* = 1.73, *M_during/late_* = 4.82, *SD_during/late_* = 1.63, *t*(4320) = 0.24, *p* = 0.81, *d* = 0.20). However, team identification and burnout were both significantly higher in countries where data were collected later in the pandemic (team identification: *M_early_* = 4.82, *SD_early_* = 1.65, *M_during/late_* = 4.99, *SD_during/late_* = 1.46, *t*(4336) = −2.35, *p* = 0.019, *d* = 0.11; burnout: *M_early_* = 3.41, *SD_early_* = 1.54, *M_during/late_* = 3.53, *SD_during/late_* = 1.65, *t*(4326) = −3.72, *p* < 0.001, *d* = 0.08). Note, though, that both effect sizes were rather small. In addition, early in the crisis, countries had a somewhat stronger effect of entrepreneurship, followed by advancement, while, later in the crisis, countries showed somewhat stronger effects of prototypicality followed by entrepreneurship.

### 3.4. Cross-Cultural Analyses (RQ2)

To explore RQ2, namely, to explore whether the results are consistent across countries with different cultural practices and beliefs, we examined the results separately within each country. This revealed a consistent pattern of support for H1 and H2 with only a few exceptions. With respect to H1, we found significant relationships between (global) identity leadership and team identification in every country, with substantial correlations ranging from *r* = 0.36 in Pakistan to *r* = 0.65 in the United States. With respect to H2, the negative indirect effect of (global) identity leadership was significant in 23 of 28 countries, while the indirect effect of dimensions of identity leadership was significant in 25 of 28 countries. Only in France no significant effects were found and there were somewhat inconsistent results in Israel and Portugal, as the indirect effect for identity leadership was not significant, whereas all indirect effects for the four dimensions were significant in both countries. In India and Slovenia, there were significant indirect effects for global identity leadership but only on two (of its four) dimensions. Among these five countries (France, Israel, Portugal, India and Slovenia) with some non-significant relationships, India and Israel included data from the early crisis, while Portugal included data from later in the crisis and France and Slovenia could not be defined as either early or late. In an additional analysis of all countries that were studied early in the crisis and those that were studied later, all indirect effects were negative and significant. Results for each country are presented in Table 5.

A further exploration of RQ2 looked at the indirect effects in the different GLOBE clusters. This inspection revealed somewhat stronger effects of identity entrepreneurship in the Anglo, Confucian Asia and Germanic European clusters, a stronger effect of identity impresarioship in Latin America, Latin Europe and Nordic Europe and a stronger effect of identity prototypicality for Southern Asia. In the Eastern European cluster, we found very similar effects of entrepreneurship, impresarioship and prototypicality.

### 3.5. Testing for Non-Linear Effects (RQ3)

In an exploratory analysis of RQ3, we tested for potential nonlinear effects. First, the regression analysis of identity leadership as predictor and team identification as criterion resulted in significant linear (*R^2^* = 0.26, *F*(1.7292) = 2609.24, *p* < 0.001) and quadratic models (*R^2^* = 0.28, *F*(1.7291) = 1383.97, *p* < 0.001). As Figure 3 shows, the association between identity leadership and team identification was especially strong at high levels of identity leadership. In other words, the higher the quality of the identity leadership they experienced, the more team members identified with their team and the lower their burnout.

In addition, we ran a non-linear analysis with MEDCURVE [53]. The quadratic effect of the mediation model was significant (*a* = 0.06, *SE* = 0.005, *t* = 10.82, *p* < 0.001) and showed that the indirect effect (H2) became stronger under increasing levels of identity leadership (3.196, 95% CI between −0.11 and −0.08; 4.757, 95% CI between −0.17 and −0.13; 6.318, 95% CI between −0.23 and −0.18). The results of this analysis are presented in Table 6.

## 4. Discussion

This research project seeks to extend our understanding of the relationship between identity leadership and employee burnout by exploring the role of team identification as a mediator of this relationship in a large multinational sample. In line with Hypothesis 1, team members’ perceptions of their supervisors’ identity leadership were associated with them identifying more strongly with their team. This relationship was stable over time and was observed in almost all national samples. In line with Hypothesis 2, the analyses testing the indirect effect of identity leadership on employee burnout via team identification found evidence of this effect in both the full dataset and in most of the individual countries, as well as across time in a German subsample. To explore Research Questions 1 and 2 further, we also compared the results to those of another large multinational sample collected in 2016/2017. These revealed broadly similar patterns of results. Whereas this supports the stability of the relationships over time and across cultures, it is noteworthy that there were some changes in the mean levels between 2016/2017 and the most recent wave collected during the COVID-19 pandemic, with comparable levels in identity leadership, but slightly higher levels of both team identification and burnout during the most recent data collection during the COVID-19 pandemic. Interestingly, such evidence that employees are more highly identified with their teams but also somewhat more burnt-out aligns with the “well-being engagement paradox” identified by Gallup in the wake of the pandemic [56].

### 4.1. Theoretical and Practical Contributions

At a theoretical level, the present research study supports claims that identity leadership has a significant role to play in the trajectory of employee burnout [29,31]. More particularly, our results accord with suggestions that identity leadership which revolves around cultivating a collective sense of “us” within teams has a bearing not only on team members’ engagement and performance but also on their well-being and stress.

A second theoretical contribution is to confirm the importance of identity leadership across time, culture and context. In this regard, identity leadership in teams appears to be relatively stable across time, with evidence of comparable mean levels in the populations that we studied across the 2016/17 and the 2020/2021 data collection points. More interestingly, the identity leadership model seems applicable across cultures in so far as we observed relatively stable patterns across diverse cultures and cultural clusters. Finally, there was little evidence that identity leadership in teams declined in the context of a global crisis (in this case, the COVID-19 pandemic). On the contrary, our results suggest that, if anything, this was a resource that increased during COVID-19 and that, thereby, helped to support and protect team members’ well-being.

A third contribution of the present research study—that has both theoretical and practical ramifications—is that we found no evidence of a “too-much-of-a-good-thing” effect whereby very high levels of identity leadership had toxic consequences for team members [45,46]. Indeed, on the contrary, our results suggested that identity leadership promoted team identification across the board and that the more identity leadership team members experienced, the more they identified with their team and the more they were thereby protected from burnout. This is a reassuring message for those looking to help leaders engage in identity leadership in the workplace (e.g., via the 5R leadership development program; see [57]) who might otherwise be concerned that the benefits of this were confined to low-intensity efforts to build and sustain social identities in the workplace.

### 4.2. Limitations and Opportunities for Future Research

The most obvious limitation of this research study is the cross-sectional nature of the data in all countries—except Germany. Furthermore, since we did not capture instrumental variables, we cannot draw causal inferences about the observed relationships. As a result, the results of our mediation analyses must be interpreted with caution. Nevertheless, we would point out that our hypotheses were derived from well-developed theorizing about leadership and employee well-being and that our findings align with the results from longitudinal and experimental studies that have isolated variables relevant to the present analysis and confirmed the robustness of relevant causal inferences [23,42,58].

Nevertheless, to address the limitations of the present research, it will certainly be important for future research projects to collect data at multiple time points and to use experimental designs to test for causal relations. There is also a need for studies that zero in on the mechanisms that serve to translate identity leadership into increased identification, thereby into better health and well-being [16,21]. In this vein, Junker and colleagues found support for a mediational chain from identification to health via support and collective efficacy [22]. Similarly, Fransen et al. provided evidence that (process-oriented) collective efficacy serves to mediate between team identification and (outcome-oriented) team confidence, suggesting that collective efficacy might play a central mediating role in supporting both performance and well-being [58]. Finally, it might be worth embracing a broader understanding of organizational identification that recognizes that this varies qualitatively not just quantitatively (e.g., in ways suggested by Kreiner and Ashforth, [59]). In line with such considerations, Ciampa and colleagues observed that ambivalent identification had distinct implications for health and this is a possibility that it would be good to investigate across time and cultures [24].

## 5. Conclusions

The present research supports the claim that leaders who build a sense of shared social identity in their teams protect team members from the adverse effects of workplace stress—specifically in the form of burnout. Our analyses also suggest that this is true over time and across diverse cultures. Finally, by providing cross-sectional evidence over two waves that team identification mediates the relationship between identity leadership and employee burnout, this study contributes to a better understanding of the central role of social identity processes in employees’ health and well-being. The bottom line here is that leadership that fosters team members’ sense of “we” and “us” is beneficial for their well-being. Moreover, since we did not find evidence for a “too-much-of-a-good-thing” effect for identity leadership (in fact, rather the opposite), it seems to be the case that the more identity leadership team members experience, the better this is for their well-being. In other words, it appears to be the case that the more leadership helps to build and consolidate a sense of “we-ness” rather than “I-ness”, the more it supports wellness rather than illness.

## Figures and Tables

**Figure 1 ijerph-18-12081-f001:**
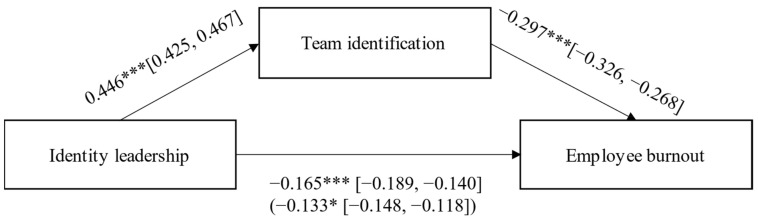
Mediation model with direct effects and the indirect effect (in parentheses) for full 2020/2021 sample. Note: * *p* < 0.05; *** *p* < 0.001.

**Figure 2 ijerph-18-12081-f002:**
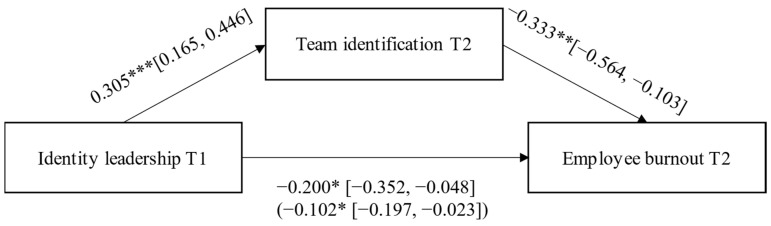
Mediation model with direct effects and the indirect effect (in parentheses) for German two-wave data. Note: * *p* < 0.05; ** *p* < 0.01; *** *p* < 0.001.

**Figure 3 ijerph-18-12081-f003:**
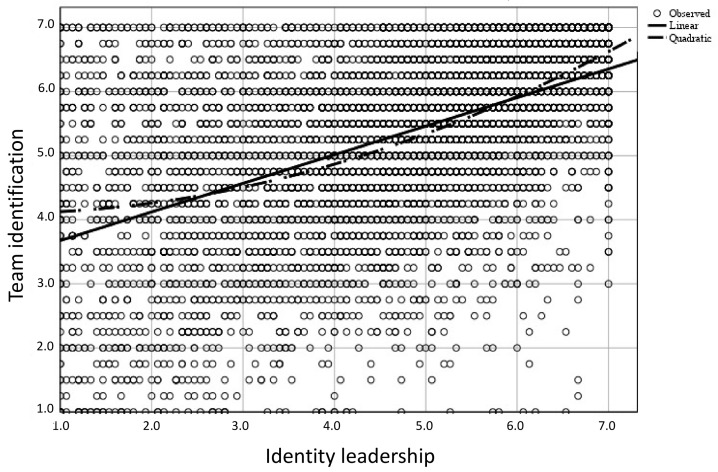
Curve estimation of the relation of identity leadership and team identification.

**Table 1 ijerph-18-12081-t001:** Sample characteristics 2021.

Nation	GLOBE Cluster	Survey Language	Participant Number	Age: % 18–25	Age: % > 55	Gender:% Female	% Leadership Responsibility	ILI Total Score
Australia	Anglo	English	269	29.4	1.9	49.8	24.2	5.0
Belgium	Germanic Europe	Dutch	285	8.8	13.7	66.0	26.7	4.7
Bosnia and Herzegovina	Eastern Europe	Bosnian	241	14.9	2.9	45.6	35.3	4.9
Brazil	Latin America	Brazilian Portuguese	222	5.9	5.9	52.3	62.2	4.5
Canada	Anglo	English	353	7.6	8.5	47.3	54.1	5.6
China	Confucian Asia	Chinese	445	14.4	1.1	45.6	58.0	5.2
France	Latin Europe	French	123	30.9	0	32.5	18.7	4.7
Germany	Germanic Europe	German	554	15.5	12.5	67.5	24.2	4.3
Greece	Eastern Europe	Greek	210	2.4	19.5	51.4	47.6	4.7
India	Southern Asia	English	192	26.6	0.5	33.3	38.0	4.9
Israel	Latin Europe	Hebrew	215	58.6	1.9	73.5		4.6
Italy	Latin Europe	Italian	191	10.5	13.1	53.4	25.1	4.0
Japan	Confucian Asia	Japanese	284	4.6	10.6	49.3	19.4	3.9
Kazakhstan	Eastern Europe	Russian	161	19.9	6.2	59.6	26.1	4.7
Netherlands	Germanic Europe	Dutch	270	14.1	17.0	50.4	25.6	4.9
Norway	Nordic Europe	Norwegian	200	1.0	18.5	37.5	33.5	4.8
Pakistan	Southern Asia	English	139	65.5	0	39.6	49.6	5.4
Pakistan	Southern Asia	Urdu	33	3.0	0	87.9	60.6	5.1
Philippines	Southern Asia	Filipino	281	24.9	4.6	64.1	44.5	5.5
Poland	Eastern Europe	Polish	375	9.9	1.3	72.8	30.7	4.3
Portugal	Latin Europe	Portuguese	202	14.4	11.4	65.3	35.6	4.7
Russia	Eastern Europe	Russian	171	1.2	9.9	81.3	59.1	4.4
Slovenia	Eastern Europe	Slovene	96	26.0	2.1	64.6	22.9	5.1
Spain	Latin Europe	Spanish	692	11.1	7.9	59.1	20.8	4.5
Switzerland	Latin Europe	English	22	13.6	0	59.1	22.7	5.2
Switzerland	Latin Europe	French	164	8.5	7.3	31.7	33.5	4.7
Switzerland	Latin Europe	German	30	3.3	10.0	30.0	40.0	5.0
Turkey	Southern Asia	Turkish	190	10.0	3.2	58.4	28.4	4.8
UK	Anglo	English	263	14.8	2.7	66.0	23.2	5.1
USA	Anglo	English	318	0.6	14.8	45.3	38.7	5.1
Uzbekistan	Eastern Europe	Russian	103	36.9	3.9	72.8	26.2	4.6
Total sample			7294	15.1	7.6	55.8	34.3	4.8

**Table 2 ijerph-18-12081-t002:** Means, standard deviations and bivariate correlations between variables 2021.

	M ^1^	SD	1	2	3	4	5	6	7
1. ILI	4.8	1.6	0.98 ^2^						
2. Prototypicality	4.8	1.6	0.94 ^3^	0.94					
3. Advancement	5.0	1.6	0.94	0.87	0.94				
4. Entrepreneurship	4.8	1.7	0.96	0.87	0.88	0.95			
5. Impresarioship	4.4	1.7	0.89	0.76	0.77	0.84	0.92		
6. Team identification	5.4	1.4	0.51	0.47	0.47	0.51	0.47	0.93	
7. Burnout	3.2	1.5	−0.32	−0.30	−0.31	−0.30	−0.26	−0.37	0.93

^1^*n* = 7294; ^2^ Cronbach’s alphas in the diagonal; ^3^ all correlations are significant with *p* < 0.001.

**Table 3 ijerph-18-12081-t003:** Means, standard deviations and bivariate correlations between variables of the German sample at T1 and T2.

	M ^1^	SD	1	2	3	4	5	6
1. ILI T1	4.4	1.8	0.98 ^2^					
2. ILI T2	4.4	1.7	0.78 **	0.98				
3. Team identification T1	5.6	1.5	0.37 **	0.24 *	0.96			
4. Team identification T2	5.6	1.4	0.40 **	0.37 **	0.73 **	0.94		
5. Burnout T1	2.7	1.3	−0.39 **	−0.42 **	−0.40 **	−0.41 **	0.92	
6. Burnout T2	2.8	1.2	−0.43 **	−0.48 **	−0.38 **	−0.48 **	0.81 **	0.92

^1^*n* = 111; ^2^ Cronbach’s alphas in the diagonal; * *p* < 0.05; ** *p* < 0.001.

**Table 4 ijerph-18-12081-t004:** Means and standard deviations of main variables in 2016 and 2021 per nation; *t*-test for independent samples.

Nation	Variable ^1^	2016 M (SD)	2021 M (SD)	Comparison of Means
Australia	ILI	4.9 (1.3)	5.0 (1.4)	*t*(578) = −1.08, *p* = 0.28, *d* = 0.07
	Team ID	5.3 (1.3)	5.3 (1.4)	*t*(578) = −0.34, *p* = 0.73, *d* = 0.00
	Burnout	3.4 (1.6)	3.6 (1.6)	*t*(578) = −1.15, *p* = 0.25, *d* = 0.13
Belgium	ILI	4.6 (1.4)	4.7 (1.5)	*t*(424) = −0.19, *p* = 0.85, *d* = 0.07
	Team ID	5.3 (1.2)	5.6 (1.2)	*t*(424) = −2.44, *p* = 0.015, *d* = 0.25
	Burnout	2.9 (1.0)	3.0 (1.3)	*t*(342) = −0.91, *p* = 0.37, *d* = 0.09
China	ILI	5.5 (1.3)	5.2 (1.1)	*t*(696) = 4.34, *p* < 0.001 *, *d* = 0.25
	Team ID	5.8 (1.2)	5.6 (1.0)	*t*(701) = 2.87, *p* = 0.004, *d* = 0.18
	Burnout	3.0 (1.5)	3.0 (1.3)	*t*(676) = 0.45, *p* = 0.66, *d* = 0.00
France	ILI	3.8 (1.6)	4.7 (1.3)	*t*(284) = −6.46, *p* < 0.001, *d* = 0.62
	Team ID	4.9 (1.6)	5.1 (1.3)	*t*(288) = −1.42, *p* = 0.16, *d* = 0.14
	Burnout	3.3 (1.4)	3.3 (1.3)	*t*(407) = −0.11, *p* = 0.92, *d* = 0.00
Germany	ILI	4.5 (1.6)	4.3 (1.7)	*t*(1004) = 2.46, *p* = 0.014, *d* = 0.12
	Team ID	5.3 (1.4)	5.5 (1.4)	*t*(1012) = −2.26, *p* = 0.024, *d* = 0.14
	Burnout	2.9 (1.4)	2.9 (1.3)	*t*(1012) = −0.35, *p* = 0.73, *d* = 0.00
Greece	ILI	4.3 (1.7)	4.7 (1.7)	*t*(479) = −2.39, *p* = 0.017, *d* = 0.24
	Team ID	4.8 (1.5)	5.3 (1.3)	*t*(471) = −4.29, *p* < 0.001 *, *d* = 0.36
	Burnout	3.6 (1.4)	3.6 (1.3)	*t*(469) = −0.31, *p* = 0.76, *d* = 0.00
India	ILI	4.8 (1.6)	4.9 (1.5)	*t*(386) = −0.65, *p* = 0.52, *d* = 0.06
	Team ID	5.3 (1.6)	5.4 (1.4)	*t*(386) = −0.28, *p* = 0.78, *d* = 0.07
	Burnout	3.0 (1.4)	3.3 (1.5)	*t*(386) = −1.90, *p* = 0.058, *d* = 0.21
Israel	ILI	4.6 (1.5)	4.6 (1.5)	*t*(521) = 0.27, *p* = 0.79, *d* = 0.00
	Team ID	---	5.2 (1.3)	*---*
	Burnout	2.9 (1.4)	3.0 (1.2)	*t*(521) = −0.63, *p* = 0.53, *d* = 0.08
Italy	ILI	4.1 (1.7)	4.0 (1.7)	*t*(358) = 0.16, *p* = 0.88, *d* = 0.06
	Team ID	4.9 (1.6)	5.0 (1.6)	*t*(358) = −0.56, *p* = 0.576, *d* = 0.06
	Burnout	3.2 (1.4)	3.4 (1.4)	*t*(358) = −1.95, *p* = 0.052, *d* = 0.14
Japan	ILI	4.1 (1.4)	3.9 (1.4)	*t*(619) = 1.63, *p* = 0.10, *d* = 0.14
	Team ID	4.1 (1.3)	4.1 (1.3)	*t*(619) = −0.283, *p* = 0.78, *d* = 0.00
	Burnout	4.1 (1.9)	3.7 (1.7)	*t*(617) = 2.71, *p* = 0.007, *d* = 0.22
Netherlands	ILI	4.8 (1.3)	4.9 (1.3)	*t*(471) = −0.83, *p* = 0.41, *d* = 0.08
	Team ID	5.4 (1.2)	5.4 (1.1)	*t*(471) = 0.36, *p* = 0.72, *d* = 0.00
	Burnout	2.8 (1.3)	2.7 (1.4)	*t*(471) = 0.11, *p* = 0.91, *d* = 0.07
Norway	ILI	4.7 (1.4)	4.8 (1.4)	*t*(527) = −1.05, *p* = 0.29, *d* = 0.07
	Team ID	5.1 (1.3)	5.3 (1.3)	*t*(527) = −1.24, *p* = 0.22, *d* = 0.15
	Burnout	3.0 (1.3)	3.0 (1.4)	*t*(527) = 0.12, *p* = 0.91, *d* = 0.00
Turkey	ILI	4.5 (1.7)	4.8 (1.6)	*t*(441) = −2.07, *p* = 0.039, *d* = 0.18
	Team ID	4.8 (1.6)	5.0 (1.5)	*t*(429) = −1.10, *p* = 0.27, *d* = 0.13
	Burnout	3.4 (1.5)	3.5 (1.6)	*t*(441) = −0.75, *p* = 0.45, *d* = 0.06

^1^*N_2016_* = 5290 for ILI and burnout, *N*_2016_ = 4982 for team identification (Team ID) because of missing values for Israel, *N*_2021_ = 7294; * after applying Bonferroni correction for multiple testing, only the two tests marked with an asterisk remain significant with *p* < 0.05.

**Table 5 ijerph-18-12081-t005:** Mediation: indirect effects on burnout via team identification for all countries and per country.

	Indirect Effect(s) of ILI (Dimensions) on Burnout			
2021—All Countries *n* = 7294	Effect	SE	95% CI	
			LL	UL
ILI	−0.13	0.01	−0.15	−0.12
Prototypicality	−0.12	0.01	−0.13	−0.11
Advancement	−0.12	0.01	−0.13	−0.11
Entrepreneurship	−0.13	0.01	−0.14	−0.11
Impresarioship	−0.12	0.01	−0.13	−0.11
2021—Early crisis *n* = 2150	Effect	SE	95% CI	
			LL	UL
ILI	−0.19	0.02	−0.23	−0.15
Prototypicality	−0.17	0.02	−0.20	−0.14
Advancement	−0.17	0.02	−0.21	−0.14
Entrepreneurship	−0.18	0.02	−0.22	−0.14
Impresarioship	−0.16	0.02	−0.20	−0.13
2021—During/late crisis *n* = 2195	Effect	SE	95% CI	
			LL	UL
ILI	−0.16	0.02	−0.19	−0.13
Prototypicality	−0.15	0.01	−0.18	−0.12
Advancement	−0.14	0.01	−0.17	−0.11
Entrepreneurship	−0.15	0.01	−0.18	−0.12
Impresarioship	−0.15	0.01	−0.17	−0.12
**Anglo**				
2021—Australia *n* = 269	Effect	SE	95% CI	
			LL	UL
ILI	−0.20	0.05	−0.31	−0.11
Prototypicality	−0.18	0.04	−0.27	−0.11
Advancement	−0.18	0.05	−0.28	−0.10
Entrepreneurship	−0.19	0.05	−0.29	−0.10
Impresarioship	−0.15	0.04	−0.23	−0.09
2021—United States *n* = 318	Effect	SE	95% CI	
			LL	UL
ILI	−0.34	0.06	−0.46	−0.23
Prototypicality	−0.31	0.06	−0.43	−0.21
Advancement	−0.32	0.05	−0.42	−0.22
Entrepreneurship	−0.33	0.05	−0.44	−0.23
Impresarioship	−0.33	0.05	−0.43	−0.24
2021—Canada *n* = 353	Effect	SE	95% CI	
			LL	UL
ILI	−0.08	0.04	−0.17	−0.01
Prototypicality	−0.07	0.04	−0.15	−0.01
Advancement	−0.08	0.04	−0.16	−0.01
Entrepreneurship	−0.07	0.04	−0.16	−0.01
Impresarioship	−0.08	0.03	−0.16	−0.02
2021—United Kingdom *n* = 263	Effect	SE	95% CI	
			LL	UL
ILI	−0.12	0.05	−0.22	−0.05
Prototypicality	−0.09	0.04	−0.18	−0.03
Advancement	−0.11	0.04	−0.20	−0.05
Entrepreneurship	−0.12	0.05	−0.22	−0.04
Impresarioship	−0.09	0.03	−0.16	−0.03
**Confucian Asia**				
2021—China *n* = 445	Effect	SE	95% CI	
			LL	UL
ILI	−0.25	0.05	−0.35	−0.16
Prototypicality	−0.18	0.03	−0.25	−0.12
Advancement	−0.23	0.04	−0.32	−0.15
Entrepreneurship	−0.24	0.05	−0.34	−0.16
Impresarioship	−0.22	0.04	−0.30	−0.15
2021—Japan *n* = 284	Effect	SE	95% CI	
			LL	UL
ILI	−0.28	0.06	−0.40	−0.16
Prototypicality	−0.26	0.05	−0.37	−0.16
Advancement	−0.25	0.05	−0.36	−0.15
Entrepreneurship	−0.26	0.06	−0.38	−0.16
Impresarioship	−0.23	0.05	−0.34	−0.13
**Eastern Europe**				
2021—Greece *n* = 210	Effect	SE	95% CI	
			LL	UL
ILI	−0.07	0.03	−0.13	−0.02
Prototypicality	−0.06	0.03	−0.12	−0.01
Advancement	−0.06	0.03	−0.12	−0.01
Entrepreneurship	−0.07	0.03	−0.13	−0.02
Impresarioship	−0.06	0.03	−0.12	−0.02
2021—Poland *n* = 375	Effect	SE	95% CI	
			LL	UL
ILI	−0.11	0.02	−0.16	−0.06
Prototypicality	−0.10	0.02	−0.14	−0.06
Advancement	−0.09	0.02	−0.14	−0.06
Entrepreneurship	−0.10	0.02	−0.15	−0.06
Impresarioship	−0.11	0.02	−0.16	−0.07
2021—Bosnia and Herzegovina *n* = 241	Effect	SE	95% CI	
			LL	UL
ILI	−0.14	0.04	−0.22	−0.07
Prototypicality	−0.13	0.03	−0.21	−0.07
Advancement	−0.13	0.04	−0.21	−0.07
Entrepreneurship	−0.14	0.04	−0.21	−0.07
Impresarioship	−0.13	0.03	−0.20	−0.07
2021—Slovenia *n* = 96	Effect	SE	95% CI	
			LL	UL
ILI	−0.06	0.05	−0.17	0.01
Prototypicality	−0.05	0.04	−0.15	0.01
Advancement	−0.04	0.04	−0.13	0.01
Entrepreneurship	−0.07	0.04	−0.16	−0.00
Impresarioship	−0.07	0.04	−0.16	−0.01
2021—Russia *n* = 171	Effect	SE	95% CI	
			LL	UL
ILI	−0.13	0.05	−0.23	−0.05
Prototypicality	−0.12	0.04	−0.20	−0.06
Advancement	−0.11	0.03	−0.18	−0.05
Entrepreneurship	−0.12	0.04	−0.20	−0.05
Impresarioship	−0.13	0.04	−0.21	−0.05
2021—Uzbekistan *n* = 103	Effect	SE	95% CI	
			LL	UL
ILI	−0.14	0.06	−0.27	−0.04
Prototypicality	−0.13	0.06	−0.26	−0.04
Advancement	−0.13	0.05	−0.25	−0.04
Entrepreneurship	−0.12	0.05	−0.24	−0.04
Impresarioship	−0.10	0.05	−0.21	−0.03
2021—Kazakhstan *n* = 161	Effect	SE	95% CI	
			LL	UL
ILI	−0.12	0.05	−0.22	−0.03
Prototypicality	−0.12	0.04	−0.21	−0.04
Advancement	−0.12	0.04	−0.21	−0.05
Entrepreneurship	−0.11	0.04	−0.20	−0.04
Impresarioship	−0.10	0.04	−0.19	−0.02
**Germanic Europe**				
2021 Belgium—*n* = 285	Effect	SE	95% CI	
			LL	UL
ILI	−0.12	0.03	−0.18	−0.06
Prototypicality	−0.11	0.03	−0.16	−0.06
Advancement	−0.10	0.03	−0.15	−0.05
Entrepreneurship	−0.12	0.03	−0.18	−0.06
Impresarioship	−0.09	0.02	−0.14	−0.05
2021—Netherlands *n* = 270	Effect	SE	95% CI	
			LL	UL
ILI	−0.20	0.05	−0.31	−0.09
Prototypicality	−0.18	0.05	−0.29	−0.08
Advancement	−0.15	0.04	−0.25	−0.07
Entrepreneurship	−0.19	0.05	−0.29	−0.09
Impresarioship	−0.17	0.04	−0.25	−0.10
2021—Germany *n* = 554	Effect	SE	95% CI	
			LL	UL
ILI	−0.08	0.02	−0.12	−0.04
Prototypicality	−0.07	0.02	−0.11	−0.04
Advancement	−0.07	0.02	−0.11	−0.04
Entrepreneurship	−0.07	0.02	−0.11	−0.04
Impresarioship	−0.08	0.02	−0.11	−0.04
**Latin America**				
2021—Brazil *n* = 222	Effect	SE	95% CI	
			LL	UL
ILI	−0.07	0.03	−0.14	−0.02
Prototypicality	−0.07	0.03	−0.13	−0.02
Advancement	−0.06	0.03	−0.12	−0.02
Entrepreneurship	−0.08	0.03	−0.14	−0.02
Impresarioship	−0.08	0.03	−0.14	−0.03
**Latin Europe**				
2021—France *n* = 123	Effect	SE	95% CI	
			LL	UL
ILI	−0.03	0.05	−0.13	0.06
Prototypicality	−0.04	0.04	−0.12	0.02
Advancement	−0.03	0.04	−0.11	0.03
Entrepreneurship	−0.04	0.05	−0.14	0.06
Impresarioship	−0.05	0.04	−0.13	0.02
2021—Italy *n* = 191	Effect	SE	95% CI	
			LL	UL
ILI	−0.14	0.03	−0.21	−0.09
Prototypicality	−0.14	0.03	−0.20	−0.08
Advancement	−0.12	0.03	−0.17	−0.07
Entrepreneurship	−0.13	0.03	−0.20	−0.08
Impresarioship	−0.15	0.03	−0.22	−0.09
2021—Portugal *n* = 202	Effect	SE	95% CI	
			LL	UL
ILI	−0.09	0.05	−0.19	0.00
Prototypicality	−0.09	0.04	−0.17	−0.01
Advancement	−0.10	0.05	−0.20	−0.01
Entrepreneurship	−0.09	0.05	−0.19	−0.01
Impresarioship	−0.09	0.04	−0.18	−0.02
2021—Switzerland *n* = 216	Effect	SE	95% CI	
			LL	UL
ILI	−0.08	0.03	−0.15	−0.03
Prototypicality	−0.06	0.03	−0.12	−0.02
Advancement	−0.08	0.03	−0.14	−0.03
Entrepreneurship	−0.07	0.03	−0.13	−0.02
Impresarioship	−0.07	0.02	−0.13	−0.03
2021—Israel *n* = 215	Effect	SE	95% CI	
			LL	UL
ILI	−0.06	0.03	−0.13	0.00
Prototypicality	−0.05	0.03	−0.11	−0.00
Advancement	−0.06	0.03	−0.12	−0.00
Entrepreneurship	−0.06	0.03	−0.13	−0.00
Impresarioship	−0.06	0.02	−0.11	−0.02
2021—Spain *n* = 692	Effect	SE	95% CI	
			LL	UL
ILI	−0.09	0.02	−0.13	−0.06
Prototypicality	−0.08	0.02	−0.12	−0.05
Advancement	−0.09	0.02	−0.12	−0.05
Entrepreneurship	−0.09	0.02	−0.13	−0.06
Impresarioship	−0.09	0.02	−0.12	−0.06
**Nordic Europe**				
2021—Norway *n* = 200	Effect	SE	95% CI	
			LL	UL
ILI	−0.13	0.06	−0.25	−0.01
Prototypicality	−0.12	0.05	−0.23	−0.03
Advancement	−0.12	0.06	−0.24	−0.00
Entrepreneurship	−0.11	0.06	−0.22	−0.01
Impresarioship	−0.12	0.05	−0.24	−0.04
**Southern Asia**				
2021—Turkey *n* = 190	Effect	SE	95% CI	
			LL	UL
ILI	−0.16	0.05	−0.26	−0.08
Prototypicality	−0.17	0.05	−0.27	−0.09
Advancement	−0.14	0.04	−0.22	−0.07
Entrepreneurship	−0.14	0.04	−0.22	−0.07
Impresarioship	−0.14	0.04	−0.22	−0.07
2021—India *n* = 192	Effect	SE	95% CI	
			LL	UL
ILI	−0.07	0.04	−0.16	0.01
Prototypicality	−0.08	0.04	−0.17	−0.01
Advancement	−0.08	0.04	−0.16	−0.02
Entrepreneurship	−0.07	0.04	−0.16	0.01
Impresarioship	−0.07	0.04	−0.15	0.00
2021—Pakistan *n* = 172	Effect	SE	95% CI	
			LL	UL
ILI	−0.09	0.04	−0.17	−0.03
Prototypicality	−0.07	0.03	−0.15	−0.02
Advancement	−0.09	0.03	−0.16	−0.03
Entrepreneurship	−0.07	0.03	−0.14	−0.02
Impresarioship	−0.06	0.03	−0.12	−0.02
2021—Philippines *n* = 281	Effect	SE	95% CI	
			LL	UL
ILI	−0.26	0.06	−0.38	−0.15
Prototypicality	−0.26	0.05	−0.37	−0.16
Advancement	−0.22	0.05	−0.32	−0.13
Entrepreneurship	−0.24	0.06	−0.36	−0.13
Impresarioship	−0.23	0.05	−0.33	−0.13

**Table 6 ijerph-18-12081-t006:** Indirect effect of X on Y through M, with a non-linear relation between X and M.

	Model Predicting Team Identification (M)	
	*Coeff*	*SE*
Constant	4.11 ***	0.09
Identity leadership (X ^1^)	−0.04	0.05
Identity leadership squared (X × X)	0.06 ***	0.01
Summary of model predicting M	*R*^2^*=* 0.28 ***	
	**Model predicting burnout (Y)**	
Constant	5.60 ***	0.07
Identity leadership (X)	−0.16 ***	0.01
Team identification (M)	−0.30 ***	0.01
Summary of model predicting Y	*R*^2^*=* 0.16 ***	
	**Θ_x_** ** ^2^ **	**95% CI**
Employees with low identity leadership (*M* = 3.20)	−0.10	−0.11–0.08
Employees with moderate identity leadership (*M* = 4.76)	−0.15	−0.17–0.13
Employees with high identity leadership (*M* = 6.32)	−0.20	−0.23–0.18

^1^ X = predictor, M = mediator, Y = criterion variable; ^2^ Θ_x_ = instantaneous indirect effect of X on Y through M at a specific value X = x; * *p* < 0.05; *** *p* < 0.001.

## Data Availability

Data are available on request to the corresponding authors.

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
