# Peer review of "Identity Leadership, Employee Burnout and the Mediating Role of Team Identification: Evidence from the Global Identity Leadership Development Project"

_ijerph, 2021, doi:10.3390/ijerph182212081_

Round 1

Reviewer 1 Report

Review comments for International Journal of Environmental Research and Public Health

Manuscript ID: ijerph-1425284

Thank you for inviting me to review this exciting manuscript. The scope of representing cultures in this dataset is impressive. Here are some of the review comments for the authors to improve the quality of this manuscript. In addition, some significant methodology issues in the manuscript need to be dealt with, which may require revision.

I. Theory and Hypotheses

a. The current theoretical framework was iterated in three parts in the manuscript: a. social identification and burnout; b. leadership and burnout; c. identity leadership, team identification, and burnout. Authors could reorganize this part according to its integrated research model. Primarily, it need to add a more substantial OB/Management literature review summary on identity leadership and its effectiveness by increasing employees’ identification toward a collective identity. Such collective identity could buffer task and emotional stresses at work to minimize burnout and promote well-being.

b. Hypotheses part – authors need to carefully reword them (i.e., indirect path H2, is it mediation or just indirect path hypothesis; which are different one and another) and link it back to the literature review part to support them. If authors also want to emphasize it as cross-cultural hypotheses, they need to be clearly stated in the hypotheses. Of course, the authors also need to provide more past cross-cultural theoretical and empirical evidence supporting the hypothesis.

Suppose authors want to explore both the universal mechanism of identity leadership and the cultural differences by comparing and contrasting. In that case, reasonable hypotheses need to be reframed in the cultural comparisons with possible past literature support. If no past empirical evidence in cultural comparison, the authors should clearly state that in the hypotheses.

In addition, testing the “too-much-of-a-good-thing” in the manuscript, was it a competing hypothesis for H1/H2? Not clear in the manuscript.

c. Other points need to be taken care of:

i. Line 19, ILI (identity leadership inventory), has not been defined anywhere before
ii. Line 21, T1& T2 dataset is not really “longitudinal”

II. Method & Results

a. Measurement Invariance (major issue)

The authors didn’t disclose any measurement invariances testing results in the manuscript. This is not reasonable for the cross-cultural hypotheses testing. In addition, measurements used in the same culture (i.e., German) at different times should also be tested for invariance. In addition, the cross-cultural comparison needs to be tested and
achieve the scalar invariance to be compared in a meaningful way. This is the major technical issue of the manuscript; the authors need to conduct the analyses and disclose the results in the manuscript.

b. Other issues
Why use the cluster methods of the “GLOBE” project? Are there any theoretical reasons? Statistical results didn’t show any cluster analyses besides the descriptive results. Hence, it is confusing to see those clusters.

Measurement reliabilities statistics didn’t report by each culture.

Missing values are using the random imputation; what is the impact on the regressional statistic analysis results?

Why use regressional analyses for the hypotheses testing? Why not using SEM?

Again, comparison of means among datasets is dangerous without testing and show robust measurement invariance statistics.

If the authors are testing the indirect path using a collective dataset, why show the results for each culture's indirect path statistics? What are the theoretical and practical reasons for doing so?

Author Response

Dear editors,

thank you very much for the fast turnaround of the paper and the positive decision to invite us for a revision. We have carefully addressed all the issues mentioned by the reviewers by rewriting parts of the paper, conducting and presenting new analyses, and adding several references. Below we specify point-by-point how we have dealt with each comment/issue.

We hope you and the reviewers find our changes satisfactory and that the paper now merits publication in the International Journal of Environmental Research and Public Health. Please let us know if there is anything else we can do at this stage.

Best wishes, the authors

Reviewer 1

Thank you for inviting me to review this exciting manuscript. The scope of representing cultures in this dataset is impressive. Here are some of the review comments for the authors to improve the quality of this manuscript. In addition, some significant methodology issues in the manuscript need to be dealt with, which may require revision.

Response: Thank you for your overall positive evaluation and for your detailed suggestions. We have taken each of your issues and recommendations seriously. We believe that they indeed have helped to strengthen the contribution of the manuscript. The additional analyses (reliabilities, invariance testing) have increased our confidence in the results.

  1. Theory and Hypotheses
  2. The current theoretical framework was iterated in three parts in the manuscript: a. social identification and burnout; b. leadership and burnout; c. identity leadership, team identification, and burnout. Authors could reorganize this part according to its integrated research model. Primarily, it need to add a more substantial OB/Management literature review summary on identity leadership and its effectiveness by increasing employees’ identification toward a collective identity. Such collective identity could buffer task and emotional stresses at work to minimize burnout and promote well-being.

Response: Thank you very much. We had, in fact, included a longer paragraph specifying exactly why social identification should buffer against stress. Maybe this was not salient enough. We state: “Speaking to these various issues, a growing body of research shows that health in the workplace is affected by the sense of identity that employees derive from their membership in social groups (i.e., their social identity) [10]. In particular, social identity researchers have argued that people’s social identities are a psychological resource and that they have important consequences for health [11, 12]. This is because, among other things, social identity is a basis for (a) the provision and receipt of social support [13], (b) a sense of connection to others [14], (c) a sense of control [15], (d) a sense of collective self-efficacy [16], and (e) a sense of meaning and purpose [17]. These processes in turn are also argued to minimise — and to help people work together to counteract — the harmful effects of various stressors they encounter in the workplace in ways that protect them from burnout [18,19].” This paragraph is followed by two paragraphs providing empirical evidence for the claims by describing four studies and one meta-analysis. We had then described in detail what identity leadership and its four components are and had included four paragraphs of prior research that leads to the statement of our hypotheses.

Nevertheless, we took your comment very seriously and reviewed the respective literature again. We included all additional studies that we could find in which identity leadership was related to team identification and measures of health (burnout, health, satisfaction, or well-being). We added several new references and also included a full new paragraph now stating: “More recently, in a sample of 363 German employees, Krug et al. also found that leaders’ identity entrepreneurship predicted well-being during the COVID-19 pandemic — specifically in the form of reduced burnout and loneliness [39]. Van Dick et al. in a survey of employees across 20 countries also found negative correlations between identity leadership and its four components and burnout and a simultaneous regression analysis showing that identity advancement was the strongest predictor of burnout. In another study of 854 Spanish employees, Laguía and colleagues found identity entrepreneurship to be positively related to positive affect and negatively related to negative affect and both types of affect in turn, related to work engagement. In the domain of sports, Fransen et al. conducted a survey study of 289 handball players and found that when they perceived their coaches, captains, and informal leaders to be strong in identity leadership, they were more identified with their teams which, in turn, increased feelings of psychological safety which was then negatively related with burnout. Finally, Steffens et al. (2014) found identity leadership related to team identification and job satisfaction in a sample of 699 US employees and simultaneous regressions revealed that identity prototypicality and identity advancement predicted job satisfaction, and identity prototypicality, identity entrepreneurship, and identity impresarioship predicted team identification.”

  1. Hypotheses part – authors need to carefully reword them (i.e., indirect path H2, is it mediation or just indirect path hypothesis; which are different one and another) and link it back to the literature review part to support them. If authors also want to emphasize it as cross-cultural hypotheses, they need to be clearly stated in the hypotheses. Of course, the authors also need to provide more past cross-cultural theoretical and empirical evidence supporting the hypothesis.

Response: Thank you for pointing us to the need of a more specific wording of the hypotheses. Following your advice, we have reformulated H2 and now state: “H2: We expect a negative indirect effect of team members’ perceptions of their supervisors’ identity leadership with team members’ burnout, via team identification.”

To be clear about our intentions, we were not following the “classic” Baron & Kenny approach in which one first tests the direct effect, then the mediator and then sees whether the direct effect fully or partially disappears which would then be named full or partial mediation. Instead, we applied a more modern but widely established approach using the PROCESS macro by Andrew Hayes (see Hayes, 2017). This “bootstrapping approach” has advantages over the classic approach as discussed, for instance by Shrout and Bolger (2002) or Hayes (2009) but also others. We agree with advocates of the “classic” mediation approach who argued that this approach should be used in truly experimental data but not with (cross-sectional) survey data. This is why we have also formulated our hypotheses following the modern approach and predict indirect effects only and not mediation. This means, we are truly interested in whether identity leadership translates into lower burnout via increasing team identification and we are not so much interested in establishing partial or full mediation. 

Shrout, P. E., & Bolger, N. (2002). Mediation in experimental and nonexperimental studies: New procedures and recommendations. Psychological Methods, 7(4), 422–445. https://doi.org/10.1037/1082-989X.7.4.422

Hayes, A. F. (2009). Beyond Baron and Kenny: Statistical mediation analysis in the new millennium. Communication Monographs76(4), 408-420.

Hayes, A. F. (2017). Introduction to mediation, moderation, and conditional process analysis: A regression-based approach. Guilford publications.

Regarding the question of cultural differences, we believe that we do not have to reformulate our two main hypotheses but that it is valid to propose testing for such differences in an exploratory way as we suggest in the (open) Research Question 2. We agree, however, that we should have been more explicit as of why we are doing so and we have added the following statement based on previous knowledge: “Testing for stabilities or differences across cultures is important as there is some previous evidence of cultural difference in the effects of identification between cultures. Lee and colleagues in a large meta-analysis with over 114 studies found stronger relationships between organizational identification and work-related attitudes and behaviors in collectivistic cultures (compared to individualistic cultures) but they did not find any other influences of uncertainty avoidance or long-term orientation.”

Lee, E.-S., Park, T.-Y., & Koo, B. (2015). Identifying organizational identification as a basis for attitudes and behaviors: A meta-analytic review. Psychological Bulletin, 141(5), 1049–1080. https://doi.org/10.1037/bul0000012

Suppose authors want to explore both the universal mechanism of identity leadership and the cultural differences by comparing and contrasting. In that case, reasonable hypotheses need to be reframed in the cultural comparisons with possible past literature support. If no past empirical evidence in cultural comparison, the authors should clearly state that in the hypotheses.

We fully agree with the core idea described in your comment. Please also see our previous comment: We have added: “Testing for stabilities or differences across cultures is important as there is some previous evidence of cultural difference in the effects of identification between cultures. Lee and colleagues in a large meta-analysis with over 114 studies found stronger relationships between organizational identification and work-related attitudes and behaviors in collectivistic cultures (compared to individualistic cultures) but they did not find any other influences of uncertainty avoidance or long-term orientation.” However, we believe that the fact that there is not enough empirical evidence across cultures for the specific relations we study in the present research does not put us in a position to reframe the hypotheses in terms of cultural comparisons. Hence, we decided to address the issue of cultural differences by posing the (open) Research Question 2. We hope that this explanation makes our decision more explicit and you agree with our assessment.

In addition, testing the “too-much-of-a-good-thing” in the manuscript, was it a competing hypothesis for H1/H2? Not clear in the manuscript.

Response: Thank you for requesting some clarification here. In the revised manuscript, we are now more specific and state: “In the present research, we were in a position to test whether identity leadership might have the same negative impact if leaders take identity-building activities to extremes (RQ3). To be clear, we were not expecting such curvilinear effects of identity leadership as we believe that there is no threshold of turning too much of good leadership into negative effects. But in the spirit of open mindedness as one of the underlying principles of good science, we put RQ3 to a test in an exploratory way.”

  1. Other points need to be taken care of:
  2. Line 19, ILI (identity leadership inventory), has not been defined anywhere before

Response: Thank you, we have corrected this and now write “identity leadership”.

  1. Line 21, T1& T2 dataset is not really “longitudinal”

Response: We are now more modest and talk about a two-wave study here and at all other places in the paper where we refer to this data and the analyses.

  1. Method & Results
  2. Measurement Invariance (major issue)

The authors didn’t disclose any measurement invariances testing results in the manuscript. This is not reasonable for the cross-cultural hypotheses testing. In addition, measurements used in the same culture (i.e., German) at different times should also be tested for invariance. In addition, the cross-cultural comparison needs to be tested and achieve the scalar invariance to be compared in a meaningful way. This is the major technical issue of the manuscript; the authors need to conduct the analyses and disclose the results in the manuscript.

Response: We had indeed not performed these analyses as we found all scales to be invariant in our earlier data set from 2016/17 and reported the results in detail in van Dick et al. (2018). We are therefore grateful for your suggestion and have now calculated invariance for all scales and found that all scales and items were indeed largely invariant. To report all the analyses and detailed results in the manuscript would have taken a lot of space, so we only provided the following summary statement in the paper:

“Before proceeding with the main analyses, we tested all scales and items for invariance across countries. Unless stated otherwise, all of the following analyses were performed with the whole dataset. For the ILI scale, the factor loadings R² and intercepts R² are good and suggest a high level of invariance of the ILI. There are 1.7% of factor loadings that are not invariant and 22.4% of intercepts that are not invariant. Averaging the proportion of non-invariant factor loadings and intercepts, the total invariance of the ILI is 12.05%, which is below the 25% threshold (Muthén & Asparouhov, 2014). For team identification, the factor loadings R² and intercepts R² are good and suggest a high level of invariance of the team identification scale. There are 0.8% of factor loadings that are not invariant and 10.8% of intercepts that are not invariant. Averaging the proportion of non-invariant factor loadings and intercepts, the total invariance of the team identification scale is 12.05%, which is below the 25% threshold. The only exception where we did not find invariance was the small subsample of 22 participants from Switzerland who answered the survey in English. For burnout, the factor loadings R² and intercepts R² are good and suggest a high level of invariance of the burnout scale. There are 5% of factor loadings that are not invariant and 47.7% of intercepts that are not invariant. Averaging the proportion of non-invariant factor loadings and intercepts, the total invariance of the burnout scale is 26.35%, which is just above the 25% threshold. This is mainly due to the small sample sizes of participants in Pakistan who have answered in Urdu (n = 33), and those in Switzerland who have answered in English (n = 22) and German (n = 30). For the longitudinal data from Germany, the factor loadings and intercepts are invariant for all items.”

Muthén, B., & Asparouhov, T. (2014). IRT studies of many groups: The alignment method. Frontiers in Psychology978, 1–7. https://doi.org/10.3389/fpsyg.2014.00978 

  1. Other issues
    Why use the cluster methods of the “GLOBE” project? Are there any theoretical reasons? Statistical results didn’t show any cluster analyses besides the descriptive results. Hence, it is confusing to see those clusters.

Response: Thank you for pointing this out. We agree that we should have explained this decision more explicitly. When one is doing cross-cultural research, one can use the collected data and explore if there are any underlying clusters. It is advisable, though, to rely on existing frameworks and the most widely used frameworks are the ones by Hofstede and the GLOBE research team. We opted for the latter framework as the GLOBE study is a) based on more recent data and b) included several countries (that we have data on) which were not part of Geert Hofstede’s studies.

Measurement reliabilities statistics didn’t report by each culture.

Response: Thank you for asking about the country-specific reliabilities. We have analyzed all scale and sub-scale reliabilities and found them very good in each case. For saving space in the paper, we have provided a summary statement and report the lowest and highest scores for each scale now. We state: “Inter-correlations of the entire sample between ILI and the four dimensions, team identification and burnout as well as the reliability of the scales are presented in Table 2. As can be seen from this table, all variables were significantly associated with each other but to a varying degree (all |r|s > .26). As can be seen, reliabilities for the full data set were excellent with Cronbach Alpha’s all exceeding .90. An inspection of the reliabilities for each country showed that there was only little variation (identity leadership: lowest alpha in Pakistan: .95, highest alpha in the United States: .98; identity prototypicality: lowest alpha in Pakistan: .82, highest alpha in the United States: .97; identity advancement: lowest alpha in Pakistan: .85, highest alpha in Bosnia and Herzegovina: .97; identity entrepreneurship: lowest alpha in Pakistan: .85, highest alpha in Norway: .97; identity impresarioship: lowest alpha in Pakistan: .86, highest alpha in Norway: .95; team identification: lowest alpha in Pakistan: .85, highest alpha in the United States: .95; burnout: lowest alpha in Greece: .88, highest alpha in the United States: .97).”

Missing values are using the random imputation; what is the impact on the regressional statistic analysis results?

Response: Thank you for this important comment. We have added more detail now and make clearer that we imputed only a very small fraction of the total data. We state: “In the overall dataset 5,234 values were imputed out of 714,812 values and therefore represents only 0.7%. Regarding the variables analyzed in this paper, 614 values were imputed out of 204,232 values which only represents 0.3%.”

We also recalculated the indirect effects with the data set without imputation and we found identical results (for instance, the indirect effect of the ILI total score on burnout via team identification was b = -.13, 95%-CI[-.15, -.12] which is perfectly identical to the second digit with the one we reported. In order to save space, we have not reported all the analyses in the paper but we added the statement: “Re-analyses of the data without imputation revealed virtually identical results.”

Why use regressional analyses for the hypotheses testing? Why not using SEM?

Response: Thank you, we believe in the spirit of parsimoniousness and that regression analysis is the simplest and appropriate method to be used here. We agree that SEM has advantages but mostly so when dealing with new constructs (for which reliability and construct validity has not been established) or with construct of different and/or low reliabilities as it accounts for measurement error. As we are using well-established constructs with excellent internal consistencies, we think that the PROCESS tool we have used here is most appropriate and provides advantages over SEM in the easy reporting of direct and indirect effects together with CIs that are easy to understand for a broad readership.

Again, comparison of means among datasets is dangerous without testing and show robust measurement invariance statistics.

Response: We agree and now have calculated invariance statistics – please see our comment above in response to your point IIa.

If the authors are testing the indirect path using a collective dataset, why show the results for each culture's indirect path statistics? What are the theoretical and practical reasons for doing so?

Response: We are now more specific and in the respective section, we have introduced our reasoning and the purposes for these tests. We now write: “To explore RQ2, namely to explore whether the results are consistent across countries with different cultural practices and beliefs, we examined the results separately within each country.”

Reviewer 2 Report

This paper presents the results of a very large global survey on identity leadership and burnout with the mediating variable of team identification. Suggest you remove the superfluous table 1 heading on page 6. It’s interesting that the data collection straddled the pandemic.

 The statistical analysis is sophisticated. The results are clear and have produced a sensible model in figure 1 with team identification as a mediator. The cross-cultural analysis shows consistent findings across many countries.

This paper achieves its aims through showing the importance of identity leadership in preventing burnout through team identification. The impact of the pandemic is well explained. Shared social identity is clearly a strong driver for reducing stress and enhancing well-being.

Author Response

Dear editors,

thank you very much for the fast turnaround of the paper and the positive decision to invite us for a revision. We have carefully addressed all the issues mentioned by the reviewers by rewriting parts of the paper, conducting and presenting new analyses, and adding several references. Below we specify point-by-point how we have dealt with each comment/issue.

We hope you and the reviewers find our changes satisfactory and that the paper now merits publication in the International Journal of Environmental Research and Public Health. Please let us know if there is anything else we can do at this stage.

Best wishes, the authors

Reviewer 2

This paper presents the results of a very large global survey on identity leadership and burnout with the mediating variable of team identification. Suggest you remove the superfluous table 1 heading on page 6. It’s interesting that the data collection straddled the pandemic.

Response: Thank you, Table 1 heading is adjusted.

 The statistical analysis is sophisticated. The results are clear and have produced a sensible model in figure 1 with team identification as a mediator. The cross-cultural analysis shows consistent findings across many countries.

Response: Thank you very much.

 This paper achieves its aims through showing the importance of identity leadership in preventing burnout through team identification. The impact of the pandemic is well explained. Shared social identity is clearly a strong driver for reducing stress and enhancing well-being.

Response: Thank you very much for your positive evaluation of our manuscript and your encouragement.